**Perspective**

# The gut feeling in motion sickness
Tessa M. W. Talsma [1] ✉ & Ksander N. de Winkel [2]

Motion sickness is a nearly universal response to the unnatural motions that are experienced when traveling by means other than the body's own faculties; in artificial realities; and in micro- and partial gravity environments. Despite being a known malady since ancient times, its underlying mechanisms, as well as the marked interpersonal variability in susceptibility, remain incompletely understood. While efferent brain-to-body signaling pathways involved in motion sickness have been previously described, recent findings on the functional role of the gut's (neuro)epithelial cells and microbiome point to a more intricate biological control system than previously appreciated. We examine (afferent) anatomical, hormonal, immune, and extracellular brain-body pathways, and their potential role in motion sickness etiology. This perspective proposes that an additional route may contribute to the pathophysiology of motion sickness, potentially under regulatory influence of the enteric nervous system. Candidate initiators, acting on these pathways, include humoral agents, enteroendocrine cells, and the gut microbiome.

Motion sickness is a malady brought about by self-motion that has affected humanity since the first use of transportation means other than the body's own faculties. Common symptoms include drowsiness, sweating, headaches, nausea, and vomiting[1]. Over the centuries, many theories have been posed on its causes. The ancient Greeks already observed that not everyone is equally susceptible, that sickness follows from body movements caused by waves, and proposed that its mechanism was an imbalance in body fluids ("humors"). The Chinese noted that children are particularly susceptible, they distinguished between effects of different modes of transportation, and suspected that agitation of the body's life force Qi lay at the heart of the syndrome[2]. Over the last century, advances in technology and understanding of our biophysiology sparked a renewed interest in the etiology of motion sickness, which has further increased with the advent of autonomous vehicles and extended reality devices.

State of the art models of motion sickness are mostly based on Oman's conceptual model[3–5], addressing motion sickness as the result of an accumulation of 'sensory conflict', which is a discrepancy between perceived motion and expected neural inputs of motion based on previous experience[6]. The model features a faster and slower "black box" process, describing the initial progression of motion sickness fairly well[5]. However, although the model can be fitted to individual data, it does not explain the vast variability in individual susceptibility, spanning several orders of magnitude[7], nor can it account for the gradual habituation observed for sea travelers[8,9] and astronauts[10] over the course of days. This suggests that there is another distinct (modulating) factor, representing a process or pathway[11,12].

Based on a synthesis of recent findings, we argue that the gut and its microbiome are likely candidates to fulfill this role, and that this could account for interpersonal differences in susceptibility. We first review the physiological system of motion sickness and the pathways in which sensory or symptom-related information travels between body and brain. We then present recent insights on the role of gut microbiome in body-to-brain signaling in scenarios of conflicting motion. Finally, we argue how the gut or its microbiome could form the 'missing link' between our current understanding of motion sickness etiology and empirical observations.

## Review
### The physiology of motion sickness
A schematic version of the brain-body relation in perception of motion and motion sickness is presented in Fig. 1. Its components are explained in the following section.

The first component is the vestibular system (Fig. 1, I). Without a functioning vestibular system, motion sickness does not occur[7,13] and sickness can occur in response to isolated vestibular stimulation[14]. The vestibular system consists of two sub-systems: The otolith organs, which respond to translational (i.e., gravitational) acceleration, and the semicircular canals, which respond to angular accelerations. Sensory signals are communicated through neural pathways and synaptic connections through the vestibular nuclei in the brainstem, where they are processed.

The visual system (Fig. 1, II) is likely to be a modulating factor in motion sickness[15,16]. Visual inputs could either exacerbate or alleviate motion sickness symptoms, depending on their alignment with vestibular signals[3]. Visually impaired people (e.g., blind) can experience motion sickness[17]. Light-sensitive cells in the retina translate optical motion stimuli into neural signals, which travel via the optic nerve to the visual cortex, where heading direction and velocity are inferred. These signals are also integrated with vestibular signals in the vestibular nuclei.

Somatosensory inputs, such as proprioceptive or haptic cues (Fig. 1, III) constitute a third component system involved in motion sickness[18]. Its

[1]Visualization Research Center (VISUS), University of Stuttgart, Stuttgart, Germany. [2]Unaffiliated: Ksander N. de Winkel. ✉e-mail: t.m.w.talsma@gmail.com

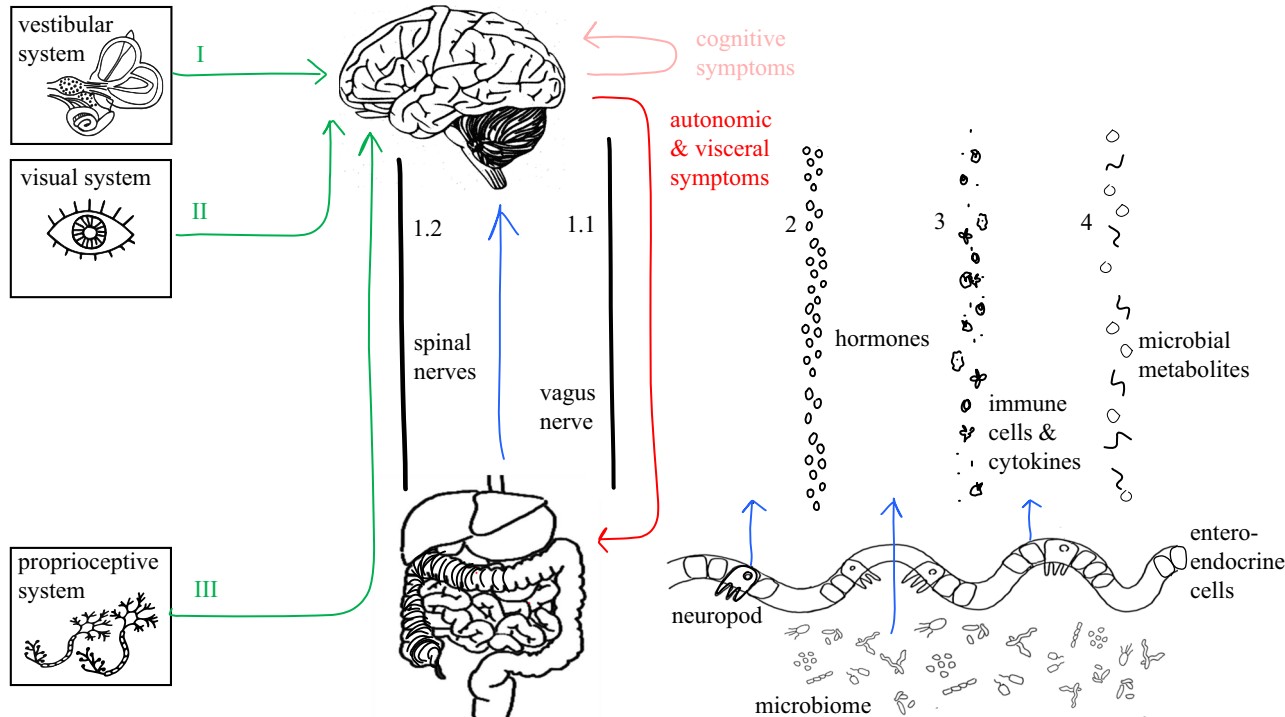

**Fig. 1 | Schematic overview of the classical physiological motion sickness model (green), the brain-to-body symptom manifestation (red) and the suggested missing body-to-brain component (blue).** From left to right: Inputs from the vestibular (I), visual (II) and proprioceptive (III) system are processed in the brain, after which symptoms (red, Table 1) are communicated over multiple autonomic routes such as the Vagus nerve (1.1). Four bidirectional brain-body pathways exist: The anatomical (1), hormonal (2), immune (3), and extracellular (4) pathway[24,25]. We posit the microbiome as potential direct and indirect (through enteroendocrine cells such as the "neuropod" cells) messengers[68]. Note: The presented pathways can overlap as complicated multifaceted interactions exist, such as hormonal components influencing the immune system.

influence, reinforcing or contradicting vestibular signals, is thought to be a secondary contributor or anticipatory factor to the vestibular and visual systems[19]. Proprioceptive cues are primarily signaled through the nerves in the spinal cord (Fig. 1, 1.2), up to the brainstem.

After integration of these inputs, the vestibular nuclei feed into five distinct networks in the brain, which are actively involved in mediation of motion sickness[20]. Increased neuronal activation (indicated by Fos protein expression) in two of these networks, correlates with severity of motion sickness symptoms and drives autonomic and gastrointestinal physiological responses (e.g., nausea, pallor and vomiting). The three other networks mediate psychological aspects (e.g., apathy, anxiety, agitation, and abulia). For details on these processing networks, see Yates et al.[21,22].

Many evident and less evident symptoms (Fig. 1, red) accompany motion sickness, further elaborated in Table 1. While, for example, the motor act of vomiting is generally well-understood, the manifestation of several other symptoms such as subjective nausea, is a complex and poorly understood interplay of nervous systems, brain regions and pathways. Symptom manifestation from mild to severe can be clustered in three categories: Cognitive response, autonomic response and visceral response[23]. While cognitive and emotional factors such as anxiety or expectation influence the perception of motion sickness symptoms, these are beyond the scope of this work.

**Brain-body pathways in motion sickness**

Classical models of motion sickness primarily emphasize brain-to-body signaling, focusing on symptom expression rather than input mechanisms. However, a clear cause-and-effect has yet to be established. Recent research highlights four bidirectional communication routes between the body and brain, that may contribute to conditions like hearing loss, tinnitus, and potentially also motion sickness[24,25]. These (1) anatomical, (2) hormonal, (3) immune, and (4) extracellular pathways are visualized in Fig. 1.

Supporting information is provided in Table 2. Our review of the evidence supports the hypothesis that peripheral (body-to-brain) signaling may play a role in the pathophysiology of motion sickness. In the following sections, we examine each pathway's evidence for afferent contributions.

**1.1 Anatomical pathway - Vagus nerve.** The Vagus nerve (Fig. 1, 1.1) is one of the significant anatomical pathways linking the brain and the body up to parts of the ear and vestibular system. It predominantly regulates parasympathetic autonomic functions like heart rate, digestion, respiration, and inflammation by transmitting signals[26], for example as a result of conflicting motion.

Contrary to its reputation as a modulating pathway with mainly efferent (brain-to-body) fibers, signals can travel bidirectionally along the nerve. The roughly 80% afferent (sensory) and 20% efferent (motor) fibers coexist within the same anatomical structure[27]. During the act of vomiting, both directions are simultaneously activated[7]. Specifically, vestibular signals and gastrointestinal signals simultaneously converge on this shared pathway, causing reciprocal influences between sensations of motion sensed by the vestibular, and gastrointestinal states such as gastric dysrhythmias[20]. Not only do both signals share the same communication pathway, the afferent Vagus signals from the gut also appear to be relayed to and processed by the same central brain region (the nucleus tractus solitarius (NTS)), which integrates vestibular information and coordinates nausea and vomiting, during, for example, conditions of sensory conflict[28]. Both inputs are then integrated and influence the brain's nausea-processing[29,30], even suggestively altering susceptibility to (motion-induced) nausea and vomiting[26]. Hereafter, the efferent Vagus (and spinal nerves) carry signals back to the body (e.g., stomach) for symptom manifestation.

Several experimental studies illustrate the critical role of Vagal afferents in motion sickness. In human studies, it is shown that targeted Vagal nerve

**Table 1 | Motion sickness responses, symptoms, related nervous systems and corresponding brain-body communication pathway**

| Response | Symptom | Nervous systems | Brain and pathways |
|---|---|---|---|
| Cognitive | Dizziness, disorientation | CNS | Vestibular signals from the inner ear travel via cranial nerves to brainstem vestibular nuclei, then project to the thalamus and cortical areas such as the parieto-insular vestibular cortex, generating spatial disorientation[18]. |
| | Fatigue, drowsiness | CNS | Vestibular nuclei influence brainstem arousal centers, reducing noradrenergic output and inducing drowsiness (i.e., sopite syndrome)[18] |
| | Headache, cognitive impairment | CNS, PSNS | Vestibular signals via the parabrachial nucleus and hypothalamus activate limbic stress circuits. Trigeminal involvement (e.g., brainstem sensitization) may explain correlation with migraine[97,98]. |
| Autonomic | Cold sweat, pallor | ANS (primarily SNS), CNS | Vestibular signals through brainstem relays (e.g., parabrachial nucleus) engage hypothalamic autonomic centers, triggering peripheral sympathetic outflow. Via thoracolumbar spinal outflow, cutaneous vasoconstriction (pallor) and stimulation of eccrine sweat glands (cold sweating)[18,99]. |
| | Heart rate, blood pressure | ANS (SNS → PSNS), Vagus Nerve | Vestibular input reaches the nucleus tractus solitarius and dorsal vagal complex. Early-phase sympathetic activation raises heart rate and blood pressure, prolonged exposure increases parasympathetic (vagal) tone, causing potential bradycardia and hypotension[26,99]. |
| Visceral | Nausea | ANS (dominantly PSNS, ENS), CNS | Vestibular signals travel to several brainstem centers (including the nucleus tractus solitarius and parabrachial nucleus) that trigger nausea[18]. These activate higher-order regions (including the insular cortex and anterior cingulate), processing interoceptive visceral sensations[18]. Onset of nausea correlates with activation of the dorsal pons and amygdala, while sustained nausea engages the insula and cingulate cortex[18]. The ENS integrates chemical and mechanical signals (e.g., luminal irritants, stretch) through intrinsic sensory neurons and local reflex circuits, modulating vagal afferent output and affecting nausea-related signaling[99,100]. |
| | Salivation, Stomach Discomfort | ANS (PSNS, ENS), CNS | Preceding vomiting, salivation and abdominal discomfort. The brainstem activates vagal efferents (via the dorsal motor nucleus of vagus) to the gut, causing altered gastric motility (including dysrhythmia and stomach upheaval)[26,32]. Simultaneously, PSNS signals via cranial nerves from brainstem salivatory nuclei, induce salivation[26]. These prodromal visceral symptoms are predominantly vagal. ENS neural circuits detect gastric distension, inflammation, and chemosensory stimuli, and modulate both motility and vagal signaling[101]. |
| | Vomiting | ANS (PSNS-dominant, ENS), CNS, Somatic motor system | Vomiting is coordinated by a central pattern generator in the medullary reticular formation, integrating inputs from the vestibular system, circulating emetic toxins, and gut afferents[18]. Once a threshold is reached, efferent signals trigger a motor response[26]: the dorsal motor nucleus of the vagus stimulates abdominal organs, while spinal motor neurons (via phrenic and intercostal nerves) induce diaphragmatic and abdominal contractions. Retrograde motility patterns and gastric muscle tone is coordinated through local motor circuits and pacemaker interactions by the ENS, contributing to the preparation for emesis[102]. |

Autonomic symptoms occur mostly by the Sympathetic Nervous System (SNS), early in the course of motion exposure, whereas as sickening motion exposure continues, the Parasympathetic Nervous System (PSNS) mostly dominates the autonomic response. Visceral symptoms relate mainly to the PSNS. For more symptoms and their organs and nervous systems involved, we refer to[11]. For interrelations with the Autonomic Nervous System (ANS) and Enteric Nervous System (ENS), see Fig. 2.

stimulation reduces motion sickness symptoms, as the artificial stimulation is said to stabilize the gut barrier function[31]. Essentially, stimulating some Vagal afferents appears to send a "calm down" signal, counteracting the aberrant firing that would induce nausea. In non-human studies, using musk shrews, it is shown that surgical disruption of Vagal signaling (vagotomy) reduced gastric rhythmic stability and abolished the normal gastric dysrhythmias induced by motion stimuli. Vagotomy did not entirely eliminate vomiting, suggesting that alternative independent pathways exist[29,32]. Similarly, in a study in which rats were rotated to induce motion sickness, vagotomy prevented conditioned taste aversion, which is an index of motion sickness[33]. Disruptions in Vagal sensory function can also exacerbate emotional responses, such as increased fear reactions in mice exposed to auditory (i.e., vestibular) stimuli[34]. These findings combined suggest that sensory conflicts involving vestibular (i.e., brain) and gastro-intestinal (i.e., body) systems might mutually modulate stress, nausea, and vomiting responses, positioning the nerve as an integrative mediator in motion sickness.

Additionally, Vagal afferents may convey gravitational sensory information from visceral organs such as the kidneys or large blood vessels to the brain. This hypothesis, by Mittelstaedt[12], arises from observations that gravity perception persists in individuals with impaired vestibular function; inertial reflexes still occurr after otolith membrane removal; and as a result of biological evidence for two distinct afferent channels signaling gravitational information. More generally, the bidirectional flow of signals between the brainstem and the body's organs over the Vagus, suggestively facilitates the integration of autonomic responses for equilibrium[35], one of the key principles during sensory conflict causing motion sickness[9]. Studies addressing conditions of altered (micro)gravity, mention visceral dynamics to significantly change as a result of the absence of gravitational forces, thereby eliminating the typical gravitational feedback of fluids in the body normally conveyed via Vagal afferents[36,37].

Whereas efferent Vagal fibers promote parasympathetic regulation as a physiological response carrier to motion-induced conflicts, aforementioned studies also indicate the role of the Vagus afferent fibres as a potential gut-derived perception carrier. While speculative, visceral organs or nearby vessels may act as additional graviceptors to the vestibular system, responding to unfamiliar altered force distributions (e.g., fluid shifts) over Vagal afferents. Such input could contribute to sensory conflict and amplify sensations of motion or imbalance.

**1.2 Anatomical pathway - Spinal cord**. Other anatomical connections related to motion sickness and vomiting are wired through the spinal cord (Fig. 1, 1.2) as part of the Central Nervous System (CNS). Vomiting is a coordinated reflex that recruits parasympathetic efferents, primarily via the Vagus nerve, alongside somatic motor output through spinal nerves, such as the phrenic nerve (to the diaphragm) and intercostal nerves (to abdominal muscles)[38]. Other (autonomic) motion sickness symptoms such as pallor (skin blood vessel constriction), cold sweating

**Table 2 | Overview of bidirectional brain-body pathways, their specific connections related to motion sickness, and the relative speed of signaling**

| Brain-body pathway | Mechanism | Speed |
|---|---|---|
| 1. Anatomical (neural) | Spinal cord nerves | Milliseconds[103] |
| | Vagus Nerve | Milliseconds to seconds[104,105], slower than spinal signaling |
| | Parasympathetic nerves (e.g., trigeminal nerve) | Seconds[106] |
| 2. Hormonal (endocrine) | Hypothalamic-pituitary-adrenal axis (cortisol), Hypothalamic-pituitary-gonadal axis (estrogen/progesterone), Arginine vasopressin, Ghrelin | Minutes to hours[107–109] |
| 3. Immune | Immune cells, cytokines, microglia, immunoglobins | Hours to days[110,111] |
| 4. Extracellular (humoral) | Microbe-derived metabolites (e.g., lactate, histamine-like) | Minutes to hours[68,112], dependent on the metabolite and microbiota composition |
| | Metabolic byproducts (e.g., glucose, serotonin) | Seconds to hours[52,113], dependent on nutritional modulation |

When considering temporal dynamics in Oman's motion sickness model[3,5], the fast and slow "black box" components would respectively match the anatomical and hormonal or humoral process.

(activation of sweat glands) and changes in heart rate or blood pressure, involve sympathetic efferents from the spinal cord[39].

Spinal afferent signals contribute to motion perception and motion sickness by relaying proprioceptive and somatosensory information about body position directly to the cerebellum and vestibular nuclei[40]. Their integration with visual and vestibular inputs is essential for accurate self-motion perception and postural control[41], although their role in triggering motion sickness through conflicting motion, is likely secondary[19].

Studies in paraplegic patients suggest that somatic graviception relies on two anatomically distinct inputs entering the spinal cord, at thoracic and cervical levels. Removal of kidneys was found to eliminate the effect of the thoracic input, implicating the kidneys in sensing gravitational forces. For the cervical input, evidence indicates that spinal nerves convey gravitational information by detecting the inertia of internal body masses[12]. While for example quadriplegics, paralyzed individuals whose proprioceptive cues cannot be communicated to the brain, do experience motion sickness[42], individuals with cervical or upper thoracic spinal cord injury often present symptoms of autonomic dysfunction similar to astronauts[43].

Clinical studies indicate that alterations in spinal afferent signaling can modify motion sickness susceptibility. For instance, patients who have impaired spinal-cerebellar connectivity, exhibit reduced susceptibility to motion sickness[44]. Conversely, abnormal cervical proprioceptive inputs, for example, as a result of chronic neck tension, are associated with increased motion sickness symptoms, suggesting that distorted spinal afferents exacerbate sensory mismatch[45]. Moreover, spinal nociceptive pathways of mice may indirectly modulate susceptibility by afferently influencing central processing of motion-related nausea[46].

Concluding, even though the spinal nerves contribute to accurate motion perception and equilibrium through afferents, and the response of symptoms such as vomiting occurs mainly through its efferents, it is questionable if this pathway is a primary determinant of individual susceptibility to motion sickness.

**2. Hormonal pathway**. The hormonal pathway (Fig. 1, 2) involves endocrine signals transmitted through the bloodstream between the brain and the body.

Motion sickness activates neuroendocrine stress responses primarily via the hypothalamic-pituitary-adrenal (HPA) axis, resulting in the release of for example corticotropin from the hypothalamus, adrenocorticotropic hormones (ACTH) from the pituitary gland, and cortisol from the adrenal glands[25,47]. Increased cortisol levels, reflecting stress-induced adrenal activation, have also been observed during motion sickness episodes[48]. Further supporting this link, individuals with primary adrenal insufficiency (i.e., impaired cortisol production) show increased motion sickness susceptibility[47]. Motion stimuli can additionally release arginine vasopressin (AVP), which is a reliable biomarker of nausea and correlates strongly with symptom severity[49,50]. Experimental infusion of AVP independently elicits subjective nausea[51], but temporal synchronization of AVP (and ACTH)

fluctuations with nausea symptoms is inconsistent, suggesting that their release represents the result of a general stress response rather than a direct nausea trigger. Motion stimuli have also been demonstrated to raise epinephrine and norepinephrine hormone levels, both of which contribute to some of the known symptoms and feed back to the brain's vestibular and autonomic centers[52]. Neuroendocrine cells in the intestinal lining additionally respond to microbial cues by secreting hormones into circulation, thereby influencing brain functions through the gut-brain axis[53], further explained in the next section. Ghrelin, a stomach-derived hormone influencing appetite and motility, was shown to correlate with autonomic symptoms and rose during seasickness[54]. As ghrelin can cross the blood-brain barrier or act on Vagal afferents, this finding suggests an afferent signaling to and influencing of the brain regions (also implicated in vomiting reflexes) in turn affecting symptom manifestation as a result of motion[55]. Increased levels of estrogen and progesterone during menstruation and pregnancy are associated with greater motion sickness susceptibility, likely due to their effects on neural processing and modulation of vasopressin and cortisol release[56].

Neurotransmitter pathways involving acetylcholine (ACh), serotonin, histamine, and substance P/neurokinin, are furthermore implicated in the mediation of motion sickness symptoms. Cholinergic (ACh) overactivity, for example, contributes significantly to motion sickness symptoms, aligning with the therapeutic effectiveness of anticholinergic medications like scopolamine[23]. Also aforementioned studies[3,11] speculate on the involvement of a cholineric component in motion sickness modulation, which will be elaborated on in the next section. Medications effective against vestibular-induced nausea, such as scopolamine, differ in effectiveness compared to those targeting other nausea forms (e.g., serotonin antagonists), implying unique or distinct neurochemical pathways for vestibular-induced sickness[23]. The existence of two effective drug categories, one blocking ACh and one activating central sympathetic areas, especially when combined, suggests competitive neural systems involved in motion sickness[57].

The role of the hormonal pathways in motion sickness can thus be represented by a systemic stress response, possibly interacting with neural circuits to modulate symptoms via the afferent (gut-to-brain) direction, particularly through the HPA axis and several peripheral hormones.

**3. Immune pathway**. The immune pathway (Fig. 1, 3) between the body and the brain encompasses signals carried by immune cells and inflammatory mediators such as cytokines and histamine. This communication is used in for example sickness during infection, but the pathway appears to play a role in motion sickness as well.

The brain can influence peripheral immune function through neurohormonal pathways. Stress induced by motion may activate immune cells in the periphery or stimulate histamine release, potentially from mast cells or through modulation by the Vagus nerve. Sympathetic nervous system activation during motion exposure can further affect immune cell trafficking and inflammatory signaling[58]. Histamine, typically associated with allergic

responses, also functions as a neurotransmitter. It has been implicated in motion-induced vomiting, with studies demonstrating that increased histamine levels worsen motion sickness symptoms in both humans[33] and animals[59]. Pharmacological interventions blocking histamine receptors or enhancing histamine degradation, effectively reduce motion sickness severity[59]. These findings suggest that mismatched vestibular input during motion may increase histaminergic activity, possibly via immune-mediated vestibular inflammation. As discussed in the previous section, antihistamines are among the most effective treatments for motion sickness[23].

Less direct but increasingly relevant evidence supports afferent immune signaling in motion sickness. Cytokines produced in for example the gastrointestinal tract, may influence the brain either through circulation or by activating Vagal afferents. One study found increased blood immunoglobulin levels after motion exposure, which correlated with symptom severity[50]. Rodent studies show that vestibular stimulation activates brain microglia and induces c-Fos expression, a marker of neuronal activation associated with vomiting, hinting at a neuroinflammatory response[59].

Additional evidence comes from clinical overlap between motion sickness and inflammatory conditions. For example, individuals suffering from migraine, which involves sterile neuroinflammation or inner ear inflammation, may show greater susceptibility to motion sickness[25]. While direct causal pathways remain under investigation, these associations suggest that immune-mediated inflammation may lower the threshold for vestibular-induced nausea.

Although specific gut-derived immune hormones directly linked to motion sickness have not yet been identified, gut-resident immune cells can release cytokines or migrate to the CNS, potentially affecting brain function. This supports the idea that the immune pathway is one of several physiological routes contributing to the onset and severity of motion sickness.

**4.1 Extracellular pathway.** The extracellular pathway (Fig. 1, 4) refers to blood-borne chemical communication from the body to the brain, encompassing shifts in pH, electrolyte balance, metabolic byproducts, and microbiome-derived metabolites. These can cross the blood-brain barrier or interact with afferent nerves, influencing brain function. This section also reviews dietary influences, which may interact with the extracellular, immune, and hormonal systems.

Prolonged nausea and reduced intake during motion sickness can cause blood glucose fluctuations. Hyperglycemia has been observed in both humans and animals, suggesting that metabolic state influences visceral symptom severity, with stable glucose levels potentially alleviating symptoms[60]. Altered blood glucose levels can affect the chemosensitivity of the area postrema and the excitability of vagal afferents, modulating nausea intensity. The area postrema, a brain region outside the blood-brain barrier, detects blood-borne signals and plays a role in nausea and vomiting responses[61]. Microbial metabolites, produced by for example gut bacteria, such as short-chain fatty acids, lactate, and neurotransmitter analogues, can enter circulation and impact brain function. The produced metabolites influence the host through a combination of immune, hormonal, and metabolic interactions. Certain gut bacterial metabolites can alter vestibular function or the threshold for nausea by affecting the vagus nerve or blood-brain barrier permeability. For example, gut microbes and their metabolites have been shown to modulate blood-brain barrier integrity and brain function[62].

**4.2 Dietary pathway.** Another form of extracellular communication is initiated by dietary intakes. Various dietary factors such as caffeine, alcohol, nicotine, and certain food constituents (e.g., histamine-rich or greasy foods) have been shown to influence susceptibility to and feelings of motion sickness. For instance, individuals prone to motion sickness are advised to avoid heavy meals and ingestion of caffeine, alcohol, and foods high in histamine content before traveling[63]. Also Vitamin C suppresses symptoms of seasickness[64] and ginger root, through mitigating excessive vagal afferent firing associated with nausea, is a classical motion sickness remedy[65]. For a comprehensive overview of nutritional influences to motion sickness, readers are referred to the review by Rahimzadeh et al.[66].

These substances affect the body through blood-borne chemicals, metabolic shifts, and microbiome interactions,

A simulator sickness study[67] found that participants experienced fewer sickness symptoms after alcohol intake. Alcohol, absorbed via the gut and crossing the blood-brain barrier, lowers vestibular fluid density, causing continuous activation of semicircular canal hair cells. This introduces vestibular noise or reduces signal reliability, potentially raising the threshold for detecting sensory mismatch and reduced symptom severity.

Graham et al.[25] conclude that dietary shifts, influencing for example the microbial populations, can impact the central auditory (i.e., vestibular) system by affecting gene expressions that target afferent neurons. Their conclusion, as well as above-mentioned findings, suggest that extracellular signals, including those from dietary intake, may affect sickness by interacting with metabolic, microbial, and neural pathways. Together, this can hint at similar mechanisms for motion-related symptoms.

## Perspective

In the previous section, we have described the influence of the mechanistic brain-body pathways in light of motion sickness, supported by theoretical and empirical evidence. According to these indicators, we suggest that signals related to motion perception and motion sickness not only travel from brain to body, but also from body to brain over one or multiple of the reviewed pathways.

According to Oman's model[3], the accumulation of symptoms over time can be described by two distinct temporal components: A fast (on the scale of seconds) and a slow component (ranging from minutes to an hour[5]). While it remains uncertain whether these correspond to separate (bio) physiological pathways, the fast component likely represents immediate neural responses, whereas the slow component has been hypothesized to reflect humoral or autonomic mechanisms[4,11]. Taken together the gradual acculumulation of sea- and space sickness over the course of days, and the speed of the body's signaling pathways (Table 2), we hypothesize that a yet unknown mechanism contributes to a third, even slower process. Emerging research suggests candidates for such afferent modulators as: (1) An unidentified cholinomimetic agent, (2) the gut's enteroendocrine and neuroepithelial sensory cells, and (3) the gut microbiome. In this section, we present these potential messengers in motion sickness physiology and propose refinements to the current model based on this body-to-brain perspective.

### The cholinomimetic agent

Sheehan et al.[11] propose that the slow component in Oman's model arises from systemic modulation by a parasympathetic cholinomimetic agent. A cholinomimetic agent is a substance that mimics the action of acetylcholine (ACh) by stimulating cholinergic receptors, thereby modulating autonomic nervous system activity through either direct receptor activation or inhibition of ACh breakdown. While the precise biochemical nature remains unidentified, it is plausible that it originates from or interacts with neural or enteroendocrine pathways, modulating cholinergic signaling at autonomic ganglia and the adrenal medulla, which in turn influences autonomic functions (e.g., gastrointestinal or cardiovascular) during motion sickness. The authors propose the yet-unidentified agent to circulate in the bloodstream and act on nicotinic ACh receptors in the body. ACh itself, which is synthesized by the parasympathetic and Enteric Nervous System (ENS) neurons, is, however, rapidly degraded and unlikely to travel far in the blood. Therefore, the messenger, representing an extracellular or neurohumoral pathway, is either more stable or indirectly influencing adrenal or autonomic responses. This generally implies a more nuanced interaction between sympathetic (i.e., fast) and parasympathetic (i.e., slow) activity and a more complex and integrated physiological response during motion sickness episodes. For further details, see Sheehan et al.[11].

### Enteroendocrine cells

Several specialized cell types located in the epithelial lining and lamina propria mediate gut-brain communication by interacting with sensory

## Box 1 | Theories on motion sickness

The underlying evolutionary cause of motion sickness remains unresolved. Many theories have been proposed throughout the years, each addressing different aspects of the phenomenon. The most widely accepted, is the Sensory Conflict Theory[6], according to which motion sickness arises from a mismatch between real-time sensory inputs and stored neural expectations based on prior experience. The Neurotoxin Theory[114] explains that symptoms mimic a poisoning response and may be an evolved defense mechanism against neurotoxins, mistakenly triggered when conflicting motion cues are interpreted as hallucinations. The Fluid Theory[115] explains the apparent sensitive frequency (around 0.2 Hz) as the natural frequency of internal fluids (e.g., blood, vestibular endolymph), where motion sickness may arise when stimuli resonating with these natural frequencies get amplified. Lastly, the Postural Instability Theory[116] posits that motion sickness results from postural instability, based on empirical observations of postural instability preceding motion sickness.

afferent nerve endings. Enterocytes (releasing cytokines), Goblet cells (influencing host-microbiome communication), Paneth cells (regulating microbial populations), and enteroendocrine cells (secreting hormones), signal directly and indirectly to the brain[68]. Recent discoveries have revealed that the intestinal lining also contains sparsely scattered sensory cells, called "neuropods"[69]. This specialized subset of enteroendocrine cells (EECs), previously known for secreting hormones in response to for example nutrients, form direct (millisecond-scale) synaptic connections specifically with vagal afferent neurons in the small intestine and colon[69,70]. Enterochromaffin (EC) cells, a subtype of EECs found in the intestines and stomach, release serotonin, which acts primarily through paracrine signaling rather than direct synaptic contact with afferent neurons[71–73]. Similar in shape to the vestibular's neuroepithelial (hair) cells, neuropods also serve as interfaces between the body's internal and external environments and the nervous system, to transmit sensed information from the outside environment. While known to signal afferently, there is no evidence to date that they participate in efferent (motor or parasympathetic) output pathways.

Neuropods detect both chemical and mechanical cues[74], such as irritants, osmolality shifts, abnormal motility, or force and temperature. As a result, they release neurotransmitters directly onto Vagal neurons which project to the NTS in brainstem circuits[75,76], involved in motion-conflict processing. In parallel, serotonin released from nearby EC cells contributes indirectly through paracrine activation of vagal afferents[72,73]. The release of neurotransmitter serotonin, for which the cells have long been implicated in nausea, is the rationale behind for example the nausea-reducing chemotherapy medicine ondansetron, attenuating Vagal afferent signals from the gut[77]. Concluding, EECs, particularly neuropods, could potentially serve both as rapid transducers of gut-derived signals to the brain during motion stimuli, contributing to the fast (i.e., neural) escalation of visceral stress or discomfort and additionally feeding into the slower, modulatory component through gradual influencing neurochemical or hormonal release.

The role of these potential messengers is illustrated in Fig. 1 (blue). For more details on these specialized epithelial cells, we refer to the work of Bohorquez and colleagues[70,74].

### The microbiome

From birth onward, the human body is inhabited by a unique community of diverse unicellular microorganisms, collectively called the microbiota. The combined (genomic) material of these organisms constitutes the microbiome. The microbiome communicates with the brain, notably through the four pathways[24] presented in Fig. 1. The gut microbiota is crucial for digestion and the maturation of the immune system of the host, establishing a reciprocal relationship by supporting immune functions while simultaneously defending against host immune responses. It triggers inflammatory responses that mobilize immune cells, indirectly influencing brain function. It produces neurotransmitters (e.g., GABA) and other psychoactive compounds, directly interacting with human cells (e.g., neuropods), in the same biochemical language[78]. Gut microbiota also influence the development and function of the ENS as microbial colonization shapes enteric neuron activity, gene expression, and gut motility. These effects appear to be reduced in germ-free mice, and restored when the microbiota is reintroduced[79]. Metabolites and endocrine factors released by gut microbiota can lastly enter the circulation, impacting neuronal function indirectly through hormonal (e.g., ghrelin) pathways. Recent studies on the gut-microbiome-brain axis highlight its broad implications for brain function and physiological processes. Some of these correlations include mood, memory, and behavioral outcomes[68,80], as well as migraine[81] and vestibular-related disorders such as hearing loss[25].

Three main indicators for microbiome involvement in motion sickness are the response to altered gravity, the response to (pharmaceutical) motion sickness remedies and the reduction of symptoms as a result of probiotic treatment. Firstly, the gut microbiota is shown to respond to stressors such as altered gravity. Exposure to microgravity can induce pathogenic microbes to become more virulent and alter their biofilm patterns[82,83]. Studies[82,84] suggested that gravitational shifts might drive the gut towards a pro-inflammatory state, as seen by several inflammation biomarkers in the form of bacterial ratios and metabolite changes. This in turn influenced neuro-vestibular health, as well as mitigating cardiovascular processes. Secondly, motion sickness medicine (e.g., cinnarizine, dimenhydrinate, and promethazine), appears to directly affect gut bacteria, suggesting that the microbiome plays a role in treatment efficacy and overall physiological responses to motion-induced stress[85]. Molefi et al.[31] showed that targeted auricular Vagus nerve stimulation, which inherently mimics microbiome effects on the Vagus nerve, significantly reduces motion sickness symptoms. Chronic auditory stress, on the other hand, alters gut microbiome composition, mediated through gut-brain signaling pathways involving Vagal afferents[86]. Thirdly, while direct empirical studies linking gut microbiota to motion sickness susceptibility are limited, two studies hint at this afferent connection. Srivastava et al.[87] studied probiotic effects on gut microbiota of subjects during a ship voyage. Probiotic consumers experienced significantly lower sea sickness incidence compared to the placebo group. The probiotic group maintained gut microbiota stability, whereas the placebo group showed significant alterations, suggesting probiotics may help preserve microbiota homeostasis under stress, potentially mitigating motion sickness. Sun et al.[88] similarly tracked sea voyage members and found that a subgroup developing persistent nausea showed distinct microbiome disruptions, with overall symptom syndromes correlating more strongly with microbiome changes than individual symptoms. Pre-voyage microbiome profiles predicted symptom development with 84% accuracy, and probiotic intervention reduced this predictive power, suggesting microbiome resilience influences vulnerability.

While abovementioned enteroendocrine (e.g., neuropod) signaling and cholinomimetic modulation are likely conserved across individuals, the microbiota composition is highly person-specific and subject to change over time. Several of these factors shaping microbial community include genetics, early-life exposures, interactions, environment, and diet[68]. This individuality means that the microbial signals influencing gut-brain communication (e.g., metabolite profiles, immune modulation, epithelial interactions) differ between people, potentially altering the strength or threshold of afferent inputs during motion exposure. We refer to extensive studies[24,68] for details on the microbiota-gut-brain axis.

## Box 2 | Nervous systems

Figure 2 presents the organization of the brain and body, i.e., the Central Nervous System (CNS) and Peripheral Nervous System (PNS) in black. The CNS encompasses the brain and spinal cord, the PNS includes all neural elements outside the CNS. The PNS(-motor branch) is subdivided into the somatic nervous system (i.e., voluntary movements) and the Autonomic Nervous System (ANS) for involuntary activity. The ANS comprises 3 main branches: Sympathetic nervous system (SNS), the parasympathetic nervous system (PSNS), and the enteric nervous system (ENS). The SNS is responsible for the "fight-or-flight" response, and the PSNS for "rest-and-digest" activities. Acute autonomic responses are typically mediated by the SNS, whereas prolonged symptoms are more closely associated with PSNS activation (see Table 1). The ENS, often referred to as the "second brain", operates semi-autonomously (red) to regulate digestive reflexes and integrate luminal and mechanical sensory input, while also maintaining bidirectional communication with the CNS, primarily via extrinsic innervation from the SNS and PSNS through the vagus and pelvic nerves. In addition to receiving extrinsic innervation, the ENS contains intestinofugal neurons, which provide a direct, gut-originating route for influencing SNS activity, and they enable long-range reflexes that bypass the CNS[117,118]. Emerging evidence suggests that the ENS contributes to afferent signaling during motion sickness by integrating local chemical, mechanical, and microbial signals and conveys them to extrinsic pathways, particularly under perturbed homeostatic conditions[119]. Given its slower processing speed and extensive crosstalk with neuroendocrine and immune systems, the ENS is a plausible mediator of an even slower modulatory component proposed in Oman's dual-process model of motion sickness[3].

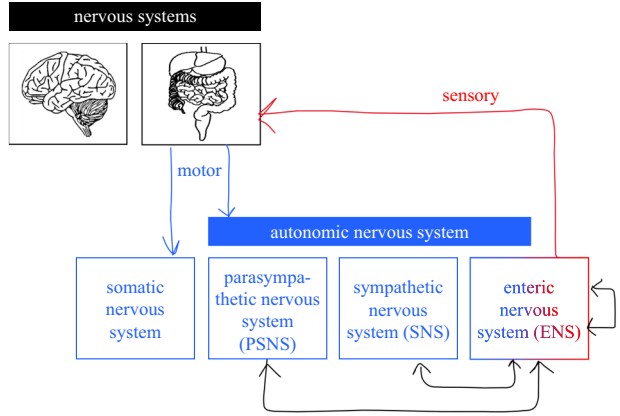

**Fig. 2 | Organization of the peripheral nervous systems and their motor (blue) and sensory (red) communication, adapted from[100].** See BOX 2 for their respective mechanisms.

### Reframing the model

As presented in Fig. 1, we suggest that the physiological model of motion sickness inputs not only signals from the vestibular system (I), the visual system (II) and the proprioceptive system (III), whereby symptoms get triggered after processing in several brain networks, but also signals from the periphery, influenced by the ENS. The ENS, often called the "second brain", is positioned to detect and respond to disturbances in internal state, relaying this information to the brain via the Vagus nerve and other aforementioned channels. The microbiome can signal to the brain, with help of for example the neuropods or ENS, and is capable of fast (i.e., neural) and slow (i.e., hormonal) signaling, potentially modulating sickness over longer time courses (i.e., immune). This modulation likely represents a third, even slower, component if seen in light of Oman's model[3].

As a result of the hypothesized connection of the gut's microbiome to motion sickness modulation, it is plausible that long-standing indicators such as the hypothesis that graviception is sensed by the viscera[12,36], a fast and slow modulatory biological component[3], and a cholinomimetic initiation[11], are not mutually exclusive.

### Discussion and future work

#### The evolutionary theory

The most widely accepted theory of motion sickness is the sensory conflict theory[6], proposing that motion sickness arises when stored expectations of motion are mismatched with real-time sensory cues. If microbiome modulation indeed contributes to motion sickness, as emerging evidence suggests, then this mechanism may have evolved to protect the host (and thereby the microbiota) by signaling environmental danger in the form of imbalance and discomfort. During motion sickness, as in other physiological stress states, both the brain and the body (including the microbiome) aim to maintain or restore internal balance, called homeostasis. One emerging idea is that the brain (supported by the CNS) and body (supported by the ENS) might detect and compare conflicting information not just spatially, but temporally, through differences in rhythmic dynamics. For example, motion-induced disruption of gastric rhythms (e.g., tachygastria), a typical consequence or driver of nausea[89], might also contribute afferently to central sensory conflict. This possibility echoes speculation by Von Gierke & Parker[36], who propose that abdominal viscera could act as independent gravity sensors. The authors suggest that at certain vibration frequencies (4-6 Hz), thoracoabdominal organs resonate, potentially sending misleading signals that conflict with vestibular cues. Moreover, studies have shown that gastric slow-wave activity of approximately 0.05 Hz[90], may phase-lock with cortical rhythms in interoceptive and sensorimotor networks[91–93], which typically operate at higher frequencies[94]. Slow-wave activity in the gut is generated by Interstitial Cells of Cajal, which act as pacemakers setting the frequency of rhythms in different gut regions. While not part of the ENS, their activity can be modulated by enteric neural input[95]. A mismatch in these rhythms, whether caused by abnormal motion or visceral over-activation, could thus introduce a new form of sensory conflict: between body (i.e., ENS) and brain (i.e., CNS) including sensory organs. Various other theories of motion sickness such as the Neurotoxin Theory (explained in BOX 1), may fit within this broader framework, rather than being mutually exclusive. Future research focusing on temporal dynamics, such as the sensitivity of specific physiological components to motion frequencies of 0.2 Hz[96], may provide important insights into the evolutionary origins and biological cause of motion sickness.

### Oman's model

This work is primarily based on Oman's model[3], with suggested adaptations[5,11], as a framework to explore mechanisms underlying motion sickness, particularly the proposed fast and slow "black boxes". While we propose a third, slower dimension, we cannot definitively assign specific processes to each box, nor include or exclude additional pathways. Moreover, Oman's model, while foundational, simplifies motion sickness to a one-dimensional nausea curve and does not fully account for symptom complexity or visual conflicts, highlighting the need for ongoing model refinements.

### Empirical validation studies

In order to test our hypothesis, we propose several experimental studies. (1) Simultaneous Electrogastrography and Electroencephalography recording

during provocative motion scenarios, which might show if gut activity shifts to co-modulate brain responses or if it precedes other symptoms. (2) Vagus Nerve stimulation frequency matching during conflicting motion, where stimulation matched (i.e., phase-locked) to gut baseline frequencies, might improve symptoms when compared to mismatched frequencies. (3) Probiotic or prebiotic intervention preceding motion sickness episodes might delay or reduce symptoms, where probiotic subjects might show reduced sickness severity as a result of increased microbial diversity as compared to baseline subjects. (4) Measurement of specific microbiota (compositions) or microbiota-released "calming" components (e.g., GABA) during conflicting motion scenarios could result in rising GABA levels with sickness severity. Similarly, the blocking GABAergic signaling would worsen symptoms. (5) Future work could also focus on disentangling which components of the microbiome are symptom-specific modulators and through which pathways they exert their influence (e.g., vagal afferents or immune or hormonal routes). For example, through specific (combinations of) motion sickness medicine in subjects targeting different processes, such as ACh or serotonin, potentially revealing which are the most prominent gut-brain pathways during motion sickness or which microbiota might be most prevalent. Symptoms of motion sickness could be measured in vagotomized patients, as well as patients with high spinal cord injury (above T6), or germ-free mice compared to colonized mice, which have not yet been linked directly to motion sickness. These experiments could ultimately support new diagnostics (e.g., microbiota-based susceptibility profiling), non-pharmacologic interventions (e.g., dietary modulation, biofeedback), or personalized treatments (e.g., Vagal modulation or targeted probiotics) to mitigate motion sickness.

## Individual differences

We propose that afferent gut-brain signaling may contribute to motion sickness susceptibility, not instead, but alongside classical mechanisms such as the involvement of the vestibular system. While our focus is on gastrointestinal factors, we acknowledge that central neural variability (e.g., in multisensory integration, mismatch thresholds), psychological or cognitive traits (e.g., anxiety), genetic and sex-related differences, prior habituation, and other peripheral factors might also influence individual responses. This perspective is not intended to exclude these, but to highlight a potentially underappreciated physiological route.

Concluding, we hypothesize that motion sickness arises not only from central sensory conflict, as proposed by the most prominent etiological account of motion sickness, but is also modulated by afferent body-to-brain signaling, originating in the gut. This suggested "missing component" in the physiology of motion sickness, likely acts via (a combination of) neural, hormonal, immune, or microbial pathways. The gut microbiome, through established communication methods with the brain, is a likely candidate of modulating motion sickness via fast (neural), slow (hormonal), and slower (immune) signaling routes. This may explain both the numerous biological and neurological correlated factors in motion sickness, as well as the gradual accumulation and eventual attenuation of symptoms observed during extended motion exposure. Recognizing the gut or enteric nervous system (i.e., "the second brain") or the microbiome as a sensory interface may explain individual variability in susceptibility. If validated by suggested future experiments, our hypothesis could pave the way for a wide range of novel interventions to mitigate motion sickness across contexts such as autonomous driving, sea travel, and simulated or virtual reality environments, situations which currently appear to affect individuals differently. Targeting body-brain homeostasis accompanied by a healthy microbiome may not only reduce sickness but also accelerate training procedures, optimize experimental outcomes, and contribute to both neuroscience and human-machine interaction advances.

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

## Author contributions

T.M.W. Talsma conceived the study and led the manuscript drafting. K.N. de Winkel contributed to writing and editing. All authors reviewed and approved the final manuscript.

## Funding

## Competing interests

The authors declare no competing interests.
