## [Transparent Peer Review file · Communications Biology]

The Gut Feeling in Motion Sickness

Corresponding Author: Ms Tessa Talsma

Version 0:

Reviewer comments:

Reviewer #1

(Remarks to the Author)

In this manuscript, Talsma and de Winkel provide an overview and an interesting perspective on the possible mechanisms underlying motion sickness. The authors propose the potential involvement of the enteric nervous system (ENS) as a novel research direction to pursue. Overall, this piece is well organised, well-written and accessible with informative summary schematics and tables. I believe will appeal to a wide readership and will be a valuable contribution to the field.

Nonetheless, there are a few areas that the authors ought to elaborate on or reconsider for a more balanced representation of the literature, particularly as it relates to neuropod cells, and also the ENS. In some instances, I am left under the impression that the authors consider the “gut”, “gut microbiome” and “ENS” as interchangeable terms, but this is not correct. The evidence provided to support their argument for possible ENS involvement is also currently lacking. The main points to be addressed are listed as follows:

1. Page 7, line 472-473: The authors state “Along the intestinal lining, including the epithelial layer and ENS...”. Since the ENS is actually not considered part of the intestinal lining itself, this sentence may be misleading. For clarity, it would be better to replace “ENS” with “sensory afferent nerve endings” which would encompass both afferent nerves of the ENS, and the vagal afferent nerves.

2. Also in ‘The neuropod cell’ section, the authors write that “This specialized subset of enteroendocrine cells, previously known for secreting hormones (e.g., ghrelin) and serotonin in response to for example nutrients, form direct (millisecond-scale) synaptic connections with afferent neurons of the Vagus nerve [84]. They do this by extending long basal projections through the lamina of the stomach and intestines, interfacing with nearby nerve terminals [85].”

Firstly, these references referring to the work of Bohorquez and colleagues relate to studies describing Cck- and Pyy-expressing enteroendocrine cells with ‘neuropods’ in small intestine and colon. The current literature provides only limited evidence for serotonin-containing enteroendocrine cells that have neuropods and make synaptic contacts with mucosal nerve endings in both the small and large intestine. It appears that 5-HT most likely acts via a paracrine action rather than through direct synaptic contact. To my knowledge, there is also no evidence for (ghrelin-containing) neuropod cells in the stomach. Otherwise, please cite the relevant literature.

Please see the following references and revise this section accordingly:

Koo A et al., 5-HT containing enteroendocrine cells characterised by morphologies, patterns of hormone co-expression, and relationships with nerve fibres in the mouse gastrointestinal tract. *Histochem Cell Biol.* 2021 Jun;155(6):623-636. doi: 10.1007/s00418-021-01972-3. Epub 2021 Feb 19. PMID: 33608804.

Spencer NJ, Kyloh MA, Travis L, Hibberd TJ. Mechanisms underlying the gut-brain communication: How enterochromaffin (EC) cells activate vagal afferent nerve endings in the small intestine. *J Comp Neurol.* 2024 Apr;532(4):e25613. doi: 10.1002/cne.25613. PMID: 38625817.

Dodds KN, Travis L, Kyloh MA, Jones LA, Keating DJ, Spencer NJ. The gut-brain axis: spatial relationship between spinal afferent nerves and 5-HT-containing enterochromaffin cells in mucosa of mouse colon. *Am J Physiol Gastrointest Liver Physiol.* 2022 May 1;322(5):G523-G533. doi: 10.1152/ajpgi.00019.2022. Epub 2022 Mar 16. PMID: 35293258.

Touhara, K.K., Rossen, N.D., Deng, F. et al. Topological segregation of stress sensors along the gut crypt–villus axis. *Nature* 640, 732–742 (2025). <https://doi.org/10.1038/s41586-024-08581-9>

3. In Box 2, the authors provide an overview of the organization of the ENS. Related to the bidirectional communication between other nervous systems, the ENS also comprises intestinofugal neurons which have projections outside the gut wall. A brief discussion on the role of intestinofugal neurons could be an interesting addition.

Please see:

Zhang T, Perkins MH, Chang H, Han W, de Araujo IE. An inter-organ neural circuit for appetite suppression. *Cell*. 2022 Jul 7;185(14):2478-2494.e28. doi: 10.1016/j.cell.2022.05.007. Epub 2022 Jun 2. PMID: 35662413; PMCID: PMC9433108.

Furness JB. Novel gut afferents: Intrinsic afferent neurons and intestinofugal neurons. *Auton Neurosci*. 2006 Apr 30;125(1-2):81-5. doi: 10.1016/j.autneu.2006.01.007. Epub 2006 Feb 14. PMID: 16476573.

4. Page 8, lines 525-527: The authors state that GABA may directly interact with human neuropod cells. Can the authors please provide supporting references for this?

5. In 'the microbiome' section on pages 8-9, the authors might also consider elaborating on their discussion as it relates to the effects of microbiota on the ENS to make a stronger link between these two. E.g. see: Obata Y et al., Neuronal programming by microbiota regulates intestinal physiology. *Nature*. 2020 Feb;578(7794):284-289. doi: 10.1038/s41586-020-1975-8. Epub 2020 Feb 5. PMID: 32025031.

6. In Box 2, it states "Emerging evidence suggests that the ENS contributes to afferent signaling during motion sickness by integrating local chemical, mechanical, and microbial inputs." It would be helpful to provide relevant references to such "emerging evidence" and to elaborate on the link to motion sickness. The authors might find this review useful: Fung, C., Vanden Berghe, P. Functional circuits and signal processing in the enteric nervous system. *Cell. Mol. Life Sci*. 77, 4505–4522 (2020). <https://doi.org/10.1007/s00018-020-03543-6>

7. Lines 599-603, the authors claim that "The microbiome, as part of the ENS specifically, can signal to the brain by means of direct and indirect (e.g., through the neuropods) communication, and is capable of fast (i.e., neural) and slow (i.e., hormonal) signaling, potentially modulating sickness over longer time courses (i.e., immune)." There are a few points that require clarification. The microbiome is not considered as "part of the ENS". The microbiota within the gut lumen is separated from the ENS and vagal nerve endings by the protective epithelial barrier/intestinal lining so they are not in direct contact. Secondly, it is unclear what the authors consider as a "direct" means of microbiome signaling to the brain given the epithelial barrier. The authors similarly mention this in their conclusion: "The gut microbiome, through both direct (e.g. vagal) and indirect (e.g. neuropod-mediated) communication with the brain..." The neuropods may indeed form synaptic contacts with vagal nerves so there is indirect communication but what is the direct pathway then? Please clarify.

8. Lines 608-609: The authors mention "indicators such as graviception sensed by the gut". It is unclear what the authors actually mean by this. What might be the mechanism? What are the receptors involved, and where within the gut might these be situated?

9. Line 636: it would be more precise to state "gastric slow-wave activity" as opposed to "gut generated slow-wave activity" as the frequency of slow-wave activity (generated by Interstitial cells of Cajal (ICCs)) can vary between different gut regions e.g. Erickson JC et al. Detection of small bowel slow-wave frequencies from noninvasive biomagnetic measurements. *IEEE Trans Biomed Eng*. 2009 Sep;56(9):2181-9. doi: 10.1109/TBME.2009.2024087. Epub 2009 Jun 2. PMID: 19497806; PMCID: PMC2760223.

Please also note that slow-wave activity in the gut is attributed to ICCs, which are not part of the ENS. Although, the ENS may modulate the activity of ICCs.

10. In Table 1, the authors list the ENS as one of the nervous systems involved in visceral responses. However, under the 'Brain and pathways' column, the enteric pathways or components involved appear lacking. Please revise.

Minor:

Page 3, line 129: spelling of dysrhythmias

Page 3, line 196: CNS not CSN

Page 4 Table 1 legend: It would be helpful to write out the abbreviations in full (e.g. SNS, PSNS...)

Page 5, Table 2: As in the previous remark, write out the abbreviations (HPA, HPG, AVP)

Page 7, line 471: spelling of 'neuropod' requires correction

Page 9, line 599: check spelling of microbiome

Page 10, line 685-686: please replace 'serotonine' with 'serotonin'

Page 10, Lines 688-689: "Symptoms of motion sickness could be measured in vasectomic patients..." I believe the authors mean to say "vagotomized" patients, unless there is also some novel link between vasectomy and motion sickness they propose to test for.

There appears to be an error with reference 11. The authors and title listed do not match. Please check.

Reviewer #2

(Remarks to the Author)

I am not convinced by the argument having read the abstract and title. However, the manuscript was a fantastic read and

provided a very compelling case for the perspective the authors put forward. I must say, it is one of the standout manuscripts in this area that I have read in the last 10 years or so, and I must congratulate the authors on this though provoking piece of work.

Only a couple of small points for the authors to consider:

1. How does this hypothesis tally with the fact that patients with bilateral vestibular failure are immune to developing the symptoms of motion sickness? This should be discussed further - if I understood correctly perhaps the vestibular input to the gut is lost?

Also please cite the paper below- as this shows experimentally how different vestibular and neurological conditions modulate motion sickness susceptibility.

J Neurol Neurosurg Psychiatry

. 2015 May;86(5):585-7. doi: 10.1136/jnnp-2014-308331. Epub 2014 Aug 11.

Motion sickness in migraine and vestibular disorders

Louisa Murdin 1, Florence Chamberlain 2, Sanjay Cheema 2, Qadeer Arshad 2, Michael A Gresty 2, John F Golding 3, Adolfo Bronstein 2

2.For the influence of cortisol/HPA axis please add the following citation:

Observational Study Exp Brain Res

. 2023 Apr;241(4):1199-1206. doi: 10.1007/s00221-023-06592-y. Epub 2023 Mar 9.

Sex-disease dimorphism underpins enhanced motion sickness susceptibility in primary adrenal insufficiency: a cross-sectional observational study

Yogan Saman # 1, Mishaal Sharif # 1, Abigail Lee 1, Shiza Ahmed 2, Ascensión Pagán 1, Maggie McGuirk 1, Oliver Rea 1, Rakesh Patel 3, Freya Bunting 1, Caitlin Spence 1, Ha-Jun Yoon 1, Elizabeta Mukaetova-Ladinska 1, Peter Rea 4, Amir Kheradmand 5, John Golding 6 7, Qadeer Arshad 8 9

Review conducted by Dr Qadeer Arshad

Associate Professor in Translational Neuroscience

Reviewer #3

(Remarks to the Author)

Talsma and de Winkel propose that motion sickness may be influenced by aspects of the gut microbiome. Given the explosion of data about and interest in the microbiome, this proposal seems worthy of consideration. Microbial involvement in motion sickness – why not? They marshal an impressive array of data on microbial processes and effects, and their proposal could help to stimulate useful new research. The principal problem with the analysis is the authors' exclusive commitment to the sensory conflict theory of motion sickness etiology. Microbial influences on motion sickness might exist and be important in ways that have nothing to do with hypothetical concepts of sensory conflict, vestibular sensitivity, neural thresholds, and so on. By maintaining their exclusive focus on sensory conflict, the authors shortchange their own proposal. Their arguments would be significantly strengthened if they were to revise to acknowledge the existence of theories of motion sickness that reject the concept of sensory conflict, such that microbial influences might be real and important but effective in very different ways.

It appears the authors may have cited a source that they have not actually read. In Box 1, they cite Riccio & Stoffregen (1991) for the postural instability theory of motion sickness etiology, and they claim that the theory “proposes that sensory conflict leads to an inability to maintain postural control”. Riccio and Stoffregen made no such claim. On the contrary, they claimed, very explicitly, that sensory conflict does not exist. Please see Riccio & Stoffregen (1991, p. 197), which includes this section heading: SENSORY CONFLICT DOES NOT EXIST. In revising, it will be essential to acknowledge that there are theories of motion sickness etiology which enjoy broad empirical support and which reject any causal role for sensory conflict. In fact, the postural instability theory proposes that motion sickness is not closely related to perceptual inputs (e.g., vestibular afferents), or to hypothetical “sensory processing”, at all.

Accurate citation of Riccio & Stoffregen (1991) should motivate the authors to consider how the gut microbiome might influence motion sickness without any appeal to hypothetical sensory conflict. It might be, for instance, that the microbiome affects movement, just as some chemicals (e.g., caffeine, alcohol) are known to influence physical movement. [In fact, the influence of “dietary factors” on motion sickness is readily accounted for in the postural instability theory but has never made much sense in the sensory conflict theory – why should caffeine (e.g.) change hypothetical discrepancies between different sensory inputs?] Stoffregen et al. (2013) provided an interpretation of seasickness that made no appeal to the concept of sensory conflict and which, instead, focused on how ship motion challenges the physical control of the body. Perhaps the authors cited this article, too, without actually reading it? For more recent statements of how postural instability might cause motion sickness with no recourse to concepts of sensory conflict, see Chang et al. (2024), Stoffregen et al. (2017), and Stanney et al. (2020; Sec. 5). Chang et al., included a detailed discussion of the issue of scientific observability in relation to both postural instability (which is directly observable) and hypothetical sensory conflict (which is not).

If hypothetical sensory conflict has no causal role in motion sickness, then it might be appropriate to reconsider the section

on The Physiology of Motion Sickness. Perhaps the physiology of motion sickness is about motor pathways, and not about sensory pathways, at all. Please note that the majority of research on the physiology of motion sickness involving “sensory pathways” has been correlational, rather than experimental. Correlation is not causation. New research might show similar (or stronger) correlations between motion sickness and activity in “motor pathways”.

The postural instability theory offers a qualitatively novel interpretation of the wide variations in individual susceptibility to motion sickness. Individuals naturally differ in motor skill, independent of any considerations of sensory issues, and so it is straightforward (in a logical sense) to suggest that individual differences in susceptibility to motion sickness might be related to individual differences in the general stability of motor control, in the speed or time course of motor adaptation to changing physical circumstances, and so on. How might any of those factors relate to the gut microbiome?

Also, it is worth noting that susceptibility varies widely within individuals but across situations: Some people who get sick in cars do not get sick at sea, or in VR, and so on. It would be helpful if the authors were to discuss how their microbial approach might (or might not) be related to these facts of the literature. Especially prominent is the widely documented fact that drivers are less susceptible than passengers (e.g., Rolnick & Lubow, 1991). Can the authors’ microbial account explain the rapid diminishment in symptoms that typically occurs when a passenger takes the wheel? Similarly, motion sickness differs between women and men, as documented in truly heroic studies with sample sizes in the thousands (e.g., Lawther & Griffin, 1988; cf. Turner & Griffin, 1999). Are there sex differences in the microbiome that might account for these effects? Sensory conflict theory is conspicuous for having little or nothing to say about those effects (see, especially, Rolnick & Lubow), which are easily accounted for in the postural instability theory (e.g., Dong et al., 2011; Koslucher et al., 2016).

Please note that “microgravity” is a misnomer, at best, and is very misleading. In Earth orbit, the magnitude of Earth’s gravitational vector is approximately 97% of its value at the Earth’s surface. Ninety-seven percent of something cannot meaningfully be described as “micro”. What is “micro” about spaceflight is the gravitoinertial force vector, which in orbit has magnitude 0. A commonly used descriptive term that is accurate with respect to the physics of spaceflight is weightlessness. People in orbit are, in physical terms, weightless. They float because they have no weight. Micro-weight (not microgravity) is a highly transient state arising from inertial (not gravitational) interactions with the interior of a spacecraft; pushing off the walls, for example. For a discussion, see Stoffregen & Riccio (1988).

The authors motivate their claims about the microbiome and motion sickness in terms of three “main indicators”: “the response to altered gravity, the response to (pharmaceutical) motion sickness remedies and the reduction of symptoms as a result of probiotic treatment”. I accept the authors’ review of the relevant microbial literature. I merely point out that none of those three effects appears to bear any inherent relation to the activity of the senses. In general, the authors’ focus on sensory sensitivity and associated neurophysiology seems to be motivated solely from their assumption of the sensory conflict theory, and not at all from their knowledge of the microbiome. In other words, the microbial effects that they review might equally be effective in relation to connections with the assembly and active control of postural movement. As one specific example, the fascinating study of Srivastava et al. (2021) does not implicate any theory of motion sickness etiology; the phrase, sensory conflict, does not appear in the article. In revising, I hope the authors will consider the addition of whatever is known about relations between the microbiome and the kinematics of human movement, independent of sensory phenomena.

Chang, C.-H., Stoffregen, T. A., Lei, M. K., Cheng, K. B., & Li, C.-C. (2024). Effects of decades of physical driving experience on pre-exposure postural precursors of motion sickness among virtual passengers. *Frontiers in Virtual Reality*, 5, 1258548. doi: 10.3389/frvir.2024.1258548

Dong, X., Yoshida, K., & Stoffregen, T. A. (2011). Control of a virtual vehicle influences postural activity and motion sickness. *Journal of Experimental Psychology: Applied*, 17, 128-138.

Koslucher, F. C., Haaland, E., & Stoffregen, T. A. (2016). Sex differences in visual performance and postural sway precede sex differences in visually induced motion sickness. *Experimental Brain Research*, 234, 313-322. 10.1007/s00221-015-4462-y

Lawther A, Griffin MJ. The motion of a ship at sea and the consequent motion sickness amongst passengers. *Ergonomics*. 1986; 29: 535 – 552 .

Lawther A, Griffin MJ. A survey of the occurrence of motion sickness amongst passengers at sea. *Aviat Space Environ Med*. 1988; 59: 399 – 406.

Rolnick, A., & Lubow, R. E. (1991). Why is the driver rarely motion sick? The role of controllability in motion sickness. *Ergonomics*, 34, 867– 879.

Stanney, K., Lawson, B. D., Rokers, B., Dennison, M., Fidopiastis, C., Stoffregen, T., Weech, S., & Fulvio, J. (2020). Identifying causes of and solutions for cybersickness in immersive technology: Reformulation of a research and development agenda. *International Journal of Human-Computer Interaction*, 36, 1783-1803. <https://doi.org/10.1080/10447318.2020.1828535>

Stoffregen, T. A., Chang, C.-H., Chen, F.-C., & Zeng, W.-J. (2017). Effects of decades of physical driving on body movement and motion sickness during virtual driving. *PLOS ONE*, 12(11): e0187120. <https://doi.org/10.1371/journal.pone.0187120>

Stoffregen, T. A., & Riccio, G. E. (1988). An ecological theory of orientation and the vestibular system. *Psychological Review*, 95, 3-14.

Turner M, Griffin MJ (1999) Motion sickness in public road transport: the relative importance of motion, vision, and individual differences. *Br J Psychol* 90:519–530.

Reviewer comments:

Reviewer #1

(Remarks to the Author)

I am pleased to say that I have no further remarks and that my concerns have been adequately addressed. Thank you to the authors for providing a well-presented and easy to follow table of all the revisions and responses to each of my comments. The manuscript was a pleasure to read.

Reviewer #2

(Remarks to the Author)

Happy with the changes made.

Reviewer #3

(Remarks to the Author)

The authors have made some changes to the manuscript in response to my original comments. However, with minimal additional changes they could increase the generality of their treatment. For example, in the paragraph beginning at Line 636, while acknowledging that the sensory conflict theory is "the most widely accepted", the authors could add (between Lines 639 and 640) something like the following:

While our view was based on Oman's model, it is neutral with respect to etiological theories of motion sickness.

In addition, Box 1 should be revised -- as I originally suggested -- to acknowledge that the postural instability theory rejects any role for sensory conflict in motion sickness etiology: Lastly, the Postural Instability Theory [41] rejects any etiological role for sensory conflict, and proposes instead that motion sickness results from unstable control of the body.

I did not request an "in depth discussion of postural instability theory". I did (and do) request a simple acknowledgement of the fact that sensory conflict may not be involved in motion sickness etiology, and that the authors' view might be correct without any dependence upon that concept. It seems to me that following my recommendations can only broaden the applicability of the authors' view.

Reviewer	Reviewer comment	Improvement (NEWLY ADDED TEXT / HIGHLIGHTED)
Reviewer #1	In this manuscript, Talsma and de Winkel provide an overview and an interesting perspective on the possible mechanisms underlying motion sickness. The authors propose the potential involvement of the enteric nervous system (ENS) as a novel research direction to pursue. Overall, this piece is well organised, well-written and accessible with informative summary schematics and tables. I believe will appeal to a wide readership and will be a valuable contribution to the field. Nonetheless, there are a few areas that the authors ought to elaborate on or reconsider for a more balanced representation of the literature, particularly as it relates to neuropod cells, and also the ENS. In some instances, I am left under the impression that the authors consider the “gut”, “gut microbiome” and “ENS” as interchangeable terms, but this is not correct. The evidence provided to support their argument for possible ENS involvement is also currently lacking. The main points to be addressed are listed as follows:	We thank the reviewer for the kind words and the helpful, constructive, expert comments as written below. The exploration of the neurobiology behind motion sickness was a relatively novel perspective for us, and we greatly appreciate corrections of any unfortunate use of terms. We have implemented all the suggestions made by the reviewer and show the changes in the following, in the column to the right of each concern. The evidence for involvement of the ENS is indeed speculative, but for example Rebollo and co-authors, in their work, provide the main argument by proving that a “novel gastric rhythm and sensory-motor processes likely interact”, also in nauseous situations. We have deleted: “While we propose a third, slower dimension initiated by the gut, we cannot definitively assign specific processes to each box, nor include or exclude additional pathways.
	1. Page 7, line 472-473: The authors state “Along the intestinal lining, including the epithelial layer and ENS...”. Since the ENS is actually not considered part of the intestinal lining itself, this sentence may be misleading. For clarity, it would be better to replace “ENS” with “sensory afferent nerve endings” which would encompass both afferent nerves of the ENS, and the vagal afferent	We have changed the sentence accordingly. p7: “Several specialized cell types located in the epithelial lining and lamina propria mediate gut-brain communication by interacting with sensory afferent nerve endings.” Section- reframing the model: “signals from the peripheral, influenced by the ENS,...” instead

	nerves.	of “peripheral, more specifically the ENS”
	2. Also in ‘The neuropod cell’ section, the authors write that “This specialized subset of enteroendocrine cells, previously known for secreting hormones (e.g., ghrelin) and serotonin in response to for example nutrients, form direct (millisecond-scale) synaptic connections with afferent neurons of the Vagus nerve [84]. They do this by extending long basal projections through the lamina of the stomach and intestines, interfacing with nearby nerve terminals [85].” Firstly, these references referring to the work of Bohorquez and colleagues relate to studies describing Cck- and Pyy-expressing enteroendocrine cells with ‘neuropods’ in small intestine and colon. The current literature provides only limited evidence for serotonin-containing enteroendocrine cells that have neuropods and make synaptic contacts with mucosal nerve endings in both the small and large intestine. It appears that 5-HT most likely acts via a paracrine action rather than through direct synaptic contact. To my knowledge, there is also no evidence for (ghrelin-containing) neuropod cells in the stomach. Otherwise, please cite the relevant literature. Please see the following references and revise this section accordingly:  - Koo A et al., 5-HT containing enteroendocrine cells characterised by morphologies, patterns of hormone co-expression, and relationships with nerve fibres in the mouse gastrointestinal tract. Histochem Cell Biol. 2021 Jun;155(6):623-636. doi: 10.1007/s00418-021-01972-3. Epub 2021 Feb 19. PMID: 33608804. - Spencer NJ, Kylloh MA, Travis L, Hibberd TJ. Mechanisms underlying the gut-brain communication: How enterochromaffin (EC) cells activate vagal afferent nerve endings in the small intestine. J Comp Neurol. 2024 Apr;532(4):e25613. doi: 10.1002/cne.25613. PMID: 38625817. - Dodds KN, Travis L, Kylloh MA, Jones LA, Keating DJ, Spencer NJ. The gut-brain axis: spatial relationship between spinal afferent nerves and 5-HT-containing enterochromaffin cells in mucosa of mouse colon. Am J Physiol Gastrointest Liver Physiol. 2022 May 1;322(5):G523-G533. doi: 10.1152/ajpgi.00019.2022. Epub 2022 Mar 16. PMID: 35293258. 	We have changed the wording so that it does not imply that these neuropods have been found in the stomach. We have also clarified the distinction between the two subsets of enteroendocrine cells (neuropods vs enterochromaffin cells), which, as far as our knowledge goes, respectively form synapses with neurons/release neurotransmitters, and act via paracrine signaling. We have revised the section (and deleted the underlined text) as follows: “This specialized subset of enteroendocrine cells (EECs) [neuropods], previously known for secreting hormones in response to for example nutrients, form direct (millisecond-scale) synaptic connections specifically with vagal afferent neurons in the small intestine and colon \cite{kaelberer-2018-gut, bohorquez2015neuroepithelial}. Enterochromaffin (EC) cells, a subtype of EECs found in the intestines and stomach, release serotonin, which acts primarily through paracrine signaling rather than direct synaptic contact with afferent neurons \cite{koo2021enteroendocrine, spencer2024gutbrain, dodds2022gut}. They do this by extending long basal projections through the lamina of the stomach and intestines, interfacing with nearby nerve terminals \cite{bohorquez2015neuroepithelial}.” And: “...In parallel, serotonin released from nearby EC cells contributes indirectly through

	 - Touhara, K.K., Rossen, N.D., Deng, F. et al. Topological segregation of stress sensors along the gut crypt–villus axis. Nature 640, 732–742 (2025). https://doi.org/10.1038/s41586-024-08581-9 	paracrine activation of vagal afferents \cite{spencer2024mechanisms, dodds2022gut}.” Small additional changes according to this information:  - abstract: “enteroendocrine cells” instead of “neuropod cells” - title of the section: “enteroendocrine cells” instead of “neuropod cell” - “Concluding, EECs, particularly neuropods, could...” - Neuropods detect both chemical and mechanical cues \cite{kaelberer-2020-neuropod}, such as irritants, osmolality shifts, abnormal gastric motility, [deleted] - microbiome section: “enteroendocrine (e.g., neuropod)” instead of “neuropod”
	3. In Box 2, the authors provide an overview of the organization of the ENS. Related to the bidirectional communication between other nervous systems, the ENS also comprises intestinofugal neurons which have projections outside the gut wall. A brief discussion on the role of intestinofugal neurons could be an interesting addition. Please see:  - Zhang T, Perkins MH, Chang H, Han W, de Araujo IE. An inter-organ neural circuit for appetite suppression. Cell. 2022 Jul 7;185(14):2478-2494.e28. doi: 10.1016/j.cell.2022.05.007. Epub 2022 Jun 2. PMID: 35662413; PMCID: PMC9433108. - Furness JB. Novel gut afferents: Intrinsic afferent neurons and intestinofugal neurons. Auton Neurosci. 2006 Apr 30;125(1-2):81-5. doi: 10.1016/j.autneu.2006.01.007. Epub 2006 Feb 14. PMID: 16476573. 	We have adjusted BOX 2 accordingly: p8: “...the SNS and PSNS through the vagus and pelvic nerves. In addition to receiving extrinsic innervation, the ENS contains intestinofugal neurons, which provide a direct, gut-originating route for influencing SNS activity, and they enable long-range reflexes that bypass the CNS (Zhang et al. (2022), Furness (2006)). Emerging evidence suggests that the ENS contributes...”
	4. Page 8, lines 525-527: The authors state that GABA may directly interact with human neuropod cells. Can the authors please provide supporting references for this?	We have identified the mistake on our side and have added reference of P. Strandwitz (2018). p8: “...It [microbiome] produces neurotransmitters (e.g., GABA) and other psychoactive compounds, directly interacting with human cells (e.g., neuropods), in the same biochemical language (Strandwitz, 2018)”
	5. In ‘the microbiome’ section on pages 8-9, the authors might also consider elaborating on their	We have added one sentence on this link as we find it complimentary to our story:

	discussion as it relates to the effects of microbiota on the ENS to make a stronger link between these two. E.g. see: Obata Y et al., Neuronal programming by microbiota regulates intestinal physiology. Nature. 2020 Feb;578(7794):284-289. doi: 10.1038/s41586-020-1975-8. Epub 2020 Feb 5. PMID: 32025031.	p8: "...biochemical language (Strandwitz, 2018). Gut microbiota also influence the development and function of the ENS as microbial colonization shapes enteric neuron activity, gene expression, and gut motility. These effects appear to be reduced in germ-free mice, and restored when the microbiota is reintroduced \cite{obata2020neuronal}. Metabolites and endocrine..."
	6. In Box 2, it states "Emerging evidence suggests that the ENS contributes to afferent signaling during motion sickness by integrating local chemical, mechanical, and microbial inputs." It would be helpful to provide relevant references to such "emerging evidence" and to elaborate on the link to motion sickness. The authors might find this review useful: Fung, C., Vanden Berghe, P. Functional circuits and signal processing in the enteric nervous system. Cell. Mol. Life Sci. 77, 4505–4522 (2020). https://doi.org/10.1007/s00018-020-03543-6	We find the suggested work indeed a very extensive review on the working principle of the ENS, we have therefore added it to our work. p8: "Emerging evidence suggests that the ENS contributes to afferent signaling during motion sickness by integrating local chemical, mechanical, and microbial signals and conveys them to extrinsic pathways, particularly under perturbed homeostatic conditions (Fung, 2020)." Further elaboration on the link to motion sickness is, in our point of view, done outside the BOX 2, in the rest of the work.
	7. Lines 599-603, the authors claim that "The microbiome, as part of the ENS specifically, can signal to the brain by means of direct and indirect (e.g., through the neuropods) communication, and is capable of fast (i.e., neural) and slow (i.e., hormonal) signaling, potentially modulating sickness over longer time courses (i.e., immune)." There are a few points that require clarification. The microbiome is not considered as "part of the ENS". The microbiota within the gut lumen is separated from the ENS and vagal nerve endings by the protective epithelial barrier/intestinal lining so they are not in direct contact. Secondly, it is unclear what the authors consider as a "direct" means of microbiome signaling to the brain given the epithelial barrier. The authors similarly mention this in their conclusion: "The gut microbiome, through both direct (e.g. vagal) and indirect (e.g. neuropod-mediated) communication with the brain..." The neuropods may indeed form synaptic contacts with vagal nerves so there is indirect communication but what is the direct pathway then? Please clarify.	We have deleted the underlined text for correctness: "...aforementioned channels. The microbiome, as part of the ENS specifically, can signal to the brain with help of for example by means of direct and indirect (e.g., through the neuropods or ENS) communication, and is capable of fast (i.e., neural) and slow (i.e., hormonal) signaling, potentially modulating sickness over longer time courses (i.e., immune)." In the conclusion, we have changed [the ENS "its"] microbiome, to "the microbiome" By "direct", we intended not through neuropods or other influenced EECs, as an extra block in an imaginary model. We see

		now that, given the epithelial barrier and the nature of gut-brain communication, the wording of “direct” and “indirect” is ambiguous. Conclusion: “The gut microbiome, through both direct (e.g., Vagal) and indirect (e.g., neuropod-mediated) established communication methods with the brain, is a likely candidate of modulating motion sickness via fast (neural), slow (hormonal) and slower (immune) signaling routes.”
	8. Lines 608-609: The authors mention “indicators such as graviception sensed by the gut”. It is unclear what the authors actually mean by this. What might be the mechanism? What are the receptors involved, and where within the gut might these be situated?	The statement “indicators such as graviception sensed by the gut”, refers to p.3 section 1.1 and 1.2 where we cite Mittelstaedt (1996) speculating the “additional graviceptor” to be located near the kidneys (potentially in the form of large blood vessels). The principle of respective authors is explained there, but, indeed, not again on p.9 as we aimed to briefly summarize all the longstanding indicators. We clarify this by a proper reference: “... it is plausible that long-standing indicators such as the hypothesis that graviception is sensed by the gut viscera (Mittelstaedt-1996, Vongierke-1994), a fast and slow modulatory biological component...”
	9. Line 636: it would be more precise to state “gastric slow-wave activity” as opposed to “gut generated slow-wave activity” as the frequency of slow-wave activity (generated by Interstitial cells of Cajal (ICCs)) can vary between different gut regions e.g. Erickson JC et al. Detection of small bowel slow-wave frequencies fom noninvasive biomagnetic measurements. IEEE Trans Biomed Eng. 2009 Sep;56(9):2181-9. doi: 10.1109/TBME.2009.2024087. Epub 2009 Jun 2. PMID: 19497806; PMCID: PMC2760223. Please also note that slow-wave activity in the gut is attributed to ICCs, which are not part of the ENS. Although, the ENS may modulate the activity of ICCs.	We have made changes accordingly: p9: “...Moreover, studies have shown that gastric slow-wave activity of approximately 0.05 Hz <code>\cite{wolpert-2020-electrogastrography}</code>, may phase-lock with cortical rhythms in interoceptive and sensorimotor networks <code>\cite{richter2017phase, rebollo-2018-stomach, rebollo2022cortical}</code>” And added: “...which typically operate at higher frequencies <code>\cite{chuang-2016-EEG}</code>. Slow-wave activity in the gut is generated by Interstitial Cells of Cajal, which act as pacemakers setting the frequency of rhythms in different gut regions. While not part of the ENS, their activity can be modulated by enteric neural input <code>\cite{erickson2009detection}</code>. A mismatch in these...”
	10. In Table 1, the authors list the ENS as one of the nervous systems involved in visceral responses. However, under the ‘Brain and	We have added the following sentences to Table 1: “The ENS integrates chemical and mechanical

	pathways' column, the enteric pathways or components involved appear lacking. Please revise.	signals (e.g., luminal irritants, stretch) through intrinsic sensory neurons and local reflex circuits, modulating vagal afferent output and affecting nausea-related signaling \cite{rao-2016-enteric,yates-1994-organization}” “ENS neural circuits detect gastric distension, inflammation, and chemosensory stimuli, and modulate both motility and vagal signaling \cite{wang2020vagal}.” “Retrograde motility patterns and gastric muscle tone is coordinated through local motor circuits and pacemaker interactions by the ENS, contributing to the preparation for emesis (Wood, 2006).”
	Minor:  [x] Page 3, line 129: spelling of dysrhythmias [x] Page 3, line 196: CNS not CSN [x] Page 4 Table 1 legend: It would be helpful to write out the abbreviations in full (e.g. SNS, PSNS...) [x] Page 5, Table 2: As in the previous remark, write out the abbreviations (HPA, HPG, AVP) [x] Page 7, line 471: spelling of 'neuropod' requires correction [x] Page 9, line 599: check spelling of microbiome [x] Page 10, line 685-686: please replace 'serotonine' with 'serotonin' [x] Page 10, Lines 688-689: “Symptoms of motion sickness could be measured in vasectomic patients...” I believe the authors mean to say “vagotomized” patients, unless there is also some novel link between vasectomy and motion sickness they propose to test for. [x] There appears to be an error with reference 11. The authors and title listed do not match. Please check. 	Thank you for the thoroughness, we have incorporated all corrections.
Reviewer #2	I am not convinced by the argument having read the abstract and title. However, the manuscript was a fantastic read and provided a very compelling case for the perspective the authors put forward. I must say, it is one of the standout manuscripts in this area that I have read in the last 10 years or so, and I must congratulate the authors on this though provoking piece of work. Only a couple of small points for the authors to consider:	We are beyond grateful for these words, especially from an expert in the field.

	1. How does this hypothesis tally with the fact that patients with bilateral vestibular failure are immune to developing the symptoms of motion sickness? This should be discussed further - if I understood correctly perhaps the vestibular input to the gut is lost?	Not necessarily because the vestibular input to the gut is lost; We still agree with references (Reason-1975, Cheung-1991, Lackner-2014, etc) stating that the vestibular system provides a necessary (perhaps even triggering) input for the manifestation of motion sickness. And, the observation that vestibular-impaired individuals do not experience sickness is not excluding/contradicting our perspective. We speculate in the discussion that a(n additional) “conflict” or modulation between the gut and the brain (+ sensory organs) could account for the conflict inducing motion sickness. To clarify, we added (p.10): “We propose that afferent gut-brain signaling may contribute to motion sickness susceptibility, not instead, but alongside classical mechanisms such as the involvement of the vestibular system.” And changed: “A mismatch in these coupled oscillations rhythms, whether caused by abnormal motion or visceral overactivation, could thus introduce a new form of sensory conflict: between body (i.e., ENS) and brain (i.e., CNS) including sensory organs.” And minor:
	Also please cite the paper below- as this shows experimentally how different vestibular and neurological conditions modulate motion sickness susceptibility. J Neurol Neurosurg Psychiatry . 2015 May;86(5):585-7. doi: 10.1136/jnnp-2014-308331. Epub 2014 Aug 11. Motion sickness in migraine and vestibular disorders Louisa Murdin 1, Florence Chamberlain 2, Sanjay Cheema 2, Qadeer Arshad 2, Michael A Gresty 2, John F Golding 3, Adolfo Bronstein 2	Citation added to Table 1 (p.4)
	2.For the influence of cortisol/HPA axis please add the following citation: Observational Study Exp Brain Res. 2023 Apr;241(4):1199-1206. doi: 10.1007/s00221-023-06592-y. Epub 2023 Mar 9. Sex-disease dimorphism underpins enhanced motion sickness susceptibility in primary adrenal insufficiency: a cross-sectional observational	Citation added to section 2-hormonal pathway (p.5): “...during motion sickness episodes \cite{meissner-2009-cortisol}. Further supporting this link, individuals with primary adrenal insufficiency (i.e., impaired cortisol production) show increased motion sickness susceptibility \cite{saman-2023-sex}. Motion stimuli can additionally...”

	study, Yougan Saman # 1, Mishaal Sharif # 1, Abigail Lee 1, Shiza Ahmed 2, Ascensión Pagán 1, Maggie McGuirk 1, Oliver Rea 1, Rakesh Patel 3, Freya Bunting 1, Caitlin Spence 1, Ha-Jun Yoon 1, Elizabeta Mukaetova-Ladinska 1, Peter Rea 4, Amir Kheradmand 5, John Golding 6 7, Qadeer Arshad 8 9	
Reviewer #3	Talsma and de Winkel propose that motion sickness may be influenced by aspects of the gut microbiome. Given the explosion of data about and interest in the microbiome, this proposal seems worthy of consideration. Microbial involvement in motion sickness – why not? They marshal an impressive array of data on microbial processes and effects, and their proposal could help to stimulate useful new research. The principal problem with the analysis is the authors' exclusive commitment to the sensory conflict theory of motion sickness etiology. Microbial influences on motion sickness might exist and be important in ways that have nothing to do with hypothetical concepts of sensory conflict, vestibular sensitivity, neural thresholds, and so on. By maintaining their exclusive focus on sensory conflict, the authors shortchange their own proposal. Their arguments would be significantly strengthened if they were to revise to acknowledge the existence of theories of motion sickness that reject the concept of sensory conflict, such that microbial influences might be real and important but effective in very different ways.	We thank the reviewer for the kind words and the different perspective. It was not our intention to exclude other theories on motion sickness than the Sensory Conflict Theory (SCT), which is the reason we included BOX1 (which is revised to correct the characterization of Postural Instability Theory, in accordance with the issue raised by the reviewer further on) SCT, along with its cybernetic and mathematical interpretations, provides a mechanistic explanation of motion sickness. Oman's model of the time course of symptom progression, which is based on SCT, was a main inspiration for the idea of an additional, slower pathway. The role of SCT (and alternatives, as mentioned in BOX1) in the narrative is otherwise minimal; limited to the introduction, and the opening of the discussion and conclusion. We have revised the statements in BOX1 and the conclusion in accordance with later comments by the reviewer, but consider an in-depth discussion of contrasts between the SCT and alternative theories undesirable: it would further complicate an already dense manuscript and is beyond the scope of our message.
	It appears the authors may have cited a source that they have not actually read. In Box 1, they cite Riccio & Stoffregen (1991) for the postural instability theory of motion sickness etiology, and they claim that the theory "proposes that sensory conflict leads to an inability to maintain postural control". Riccio and Stoffregen made no such claim. On the contrary, they claimed, very explicitly, that sensory conflict does not exist. Please see Riccio & Stoffregen (1991, p. 197), which includes this section heading: SENSORY CONFLICT DOES NOT EXIST. In revising, it will be essential to acknowledge that there are theories of motion sickness etiology which enjoy	We are well aware of the PIT and its rejection of the concept of sensory conflict, and we regret the unfortunate mix-up in BOX1. We have revised it to express the reviewers' point: Lastly, the Postural Instability Theory [41], but instead posits that motion sickness results from postural instability, based on empirical observations of postural instability preceding motion sickness. In addition, we have toned down the opening of the conclusion such that it no longer suggests that all accounts of motion sickness

	broad empirical support and which reject any causal role for sensory conflict. In fact, the postural instability theory proposes that motion sickness is not closely related to perceptual inputs (e.g., vestibular afferents), or to hypothetical “sensory processing”), at all.	rely on the concept of sensory conflict. Concluding, we hypothesize that motion sickness arises not only from central sensory conflict, as proposed by the most prominent etiological account of motion sickness, but is also modulated by afferent body-to-brain signaling, originating in the gut.
	Accurate citation of Riccio & Stoffregen (1991) should motivate the authors to consider how the gut microbiome might influence motion sickness without any appeal to hypothetical sensory conflict. It might be, for instance, that the microbiome affects movement, just as some chemicals (e.g., caffeine, alcohol) are known to influence physical movement. [In fact, the influence of “dietary factors” on motion sickness is readily accounted for in the postural instability theory but has never made much sense in the sensory conflict theory – why should caffeine (e.g.) change hypothetical discrepancies between different sensory inputs?] Stoffregen et al. (2013) provided an interpretation of seasickness that made no appeal to the concept of sensory conflict and which, instead, focused on how ship motion challenges the physical control of the body. Perhaps the authors cited this article, too, without actually reading it? For more recent statements of how postural instability might cause motion sickness with no recourse to concepts of sensory conflict, see Chang et al. (2024), Stoffregen et al. (2017), and Stanney et al. (2020; Sec. 5). Chang et al., included a detailed discussion of the issue of scientific observability in relation to both postural instability (which is directly observable) and hypothetical sensory conflict (which is not).	We have revised the citation of Riccio & Stoffregen. As argued above, an in-depth discussion of the PIT is beyond the scope of our manuscript.
	If hypothetical sensory conflict has no causal role in motion sickness, then it might be appropriate to reconsider the section on The Physiology of Motion Sickness. Perhaps the physiology of motion sickness is about motor pathways, and not about sensory pathways, at all. Please note that the majority of research on the physiology of motion sickness involving “sensory pathways” has been correlational, rather than experimental. Correlation is not causation. New research might show similar (or stronger) correlations between motion sickness and activity in “motor pathways”.	

	The postural instability theory offers a qualitatively novel interpretation of the wide variations in individual susceptibility to motion sickness. Individuals naturally differ in motor skill, independent of any considerations of sensory issues, and so it is straightforward (in a logical sense) to suggest that individual differences in susceptibility to motion sickness might be related to individual differences in the general stability of motor control, in the speed or time course of motor adaptation to changing physical circumstances, and so on. How might any of those factors relate to the gut microbiome?	
	Also, it is worth noting that susceptibility varies widely within individuals but across situations: Some people who get sick in cars do not get sick at sea, or in VR, and so on. It would be helpful if the authors were to discuss how their microbial approach might (or might not) be related to these facts of the literature.	Our aim for the manuscript is simply to highlight the potential role of the gut. Differential effects of situations will need to be addressed in future research. We have added the following to the discussion on page 10: If validated by suggested future experiments, our hypothesis could pave the way for a wide range of novel interventions to mitigate motion sickness across contexts such as autonomous driving, sea travel, and simulated or virtual reality environments, situations which currently appear to affect individuals differently. and: While our focus is on gastrointestinal factors, we acknowledge that central neural variability (e.g., in multisensory integration, mismatch thresholds), psychological or cognitive traits (e.g., anxiety), genetic and sex-related differences, prior habituation and other peripheral factors might also influence individual responses. This perspective is not intended to exclude these, but to highlight a potentially underappreciated physiological route.
	Especially prominent is the widely documented fact that drivers are less susceptible than passengers (e.g., Rolnick & Lubow, 1991). Can the authors' microbial account explain the rapid diminishment in symptoms that typically occurs when a passenger takes the wheel? Similarly, motion sickness differs between women and men, as documented in truly heroic studies with sample sizes in the thousands (e.g., Lawther & Griffin, 1988; cf. Turner & Griffin, 1999). Are there sex differences in the microbiome that might account for these effects?	We argue that the microbial account could form a slower pathway. Hence it is not a plausible candidate to account for different driver/passenger susceptibilities.

	Sensory conflict theory is conspicuous for having little or nothing to say about those effects (see, especially, Rolnick & Lubow), which are easily accounted for in the postural instability theory (e.g., Dong et al., 2011; Koslucher et al., 2016).	
	Please note that “microgravity” is a misnomer, at best, and is very misleading. In Earth orbit, the magnitude of Earth’s gravitational vector is approximately 97% of its value at the Earth’s surface. Ninety-seven percent of something cannot meaningfully be described as “micro”. What is “micro” about spaceflight is the gravito-inertial force vector, which in orbit has magnitude 0. A commonly used descriptive term that is accurate with respect to the physics of spaceflight is weightlessness. People in orbit are, in physical terms, weightless. They float because they have no weight. Micro-weight (not microgravity) is a highly transient state arising from inertial (not gravitational) interactions with the interior of a spacecraft; pushing off the walls, for example. For a discussion, see Stoffregen & Riccio (1988).	We had never considered that the term microgravity could be misleading, and appreciate the notion. However, the terminology is widely used and consistent with the cited work in relation to what we address. For instance, respective studies also name it that; “microgravity” [Ibrahim et al. (2024), Turrone et al. (2020), Oman et al...]. Furthermore, the Cambridge dictionary, as well as Wikipedia presents weightlessness and zero gravity as synonyms of microgravity [https://simple.wikipedia.org/wiki/Microgravity]. We believe that changing the term would be potentially confusing.
	The authors motivate their claims about the microbiome and motion sickness in terms of three “main indicators”: “the response to altered gravity, the response to (pharmaceutical) motion sickness remedies and the reduction of symptoms as a result of probiotic treatment”. I accept the authors’ review of the relevant microbial literature. I merely point out that none of those three effects appears to bear any inherent relation to the activity of the senses. In general, the authors’ focus on sensory sensitivity and associated neurophysiology seems to be motivated solely from their assumption of the sensory conflict theory, and not at all from their knowledge of the microbiome. In other words, the microbial effects that they review might equally be effective in relation to connections with the assembly and active control of postural movement. As one specific example, the fascinating study of Srivastava et al. (2021) does not implicate any theory of motion sickness etiology; the phrase, sensory conflict, does not appear in the article. In revising, I hope the authors will consider the addition of whatever is known about relations between the microbiome and the kinematics of human movement, independent of sensory phenomena.	
	 ● Chang, C.-H., Stoffregen, T. A., Lei, M. K., 	We thank the reviewer for the extensive list of

	Cheng, K. B., & Li, C.-C. (2024). Effects of decades of physical driving experience on pre-exposure postural precursors of motion sickness among virtual passengers. Frontiers in Virtual Reality, 5, 1258548. doi: 10.3389/frvir.2024.1258548  ● Dong, X., Yoshida, K., & Stoffregen, T. A. (2011). Control of a virtual vehicle influences postural activity and motion sickness. Journal of Experimental Psychology: Applied, 17, 128-138. ● Koslucher, F. C., Haaland, E., & Stoffregen, T. A. (2016). Sex differences in visual performance and postural sway precede sex differences in visually induced motion sickness. Experimental Brain Research, 234, 313-322. 10.1007/s00221-015-4462-y ● Lawther A, Griffin MJ. The motion of a ship at sea and the consequent motion sickness amongst passengers. Ergonomics. 1986; 29: 535 – 552 . ● Lawther A, Griffin MJ. A survey of the occurrence of motion sickness amongst passengers at sea. Aviat Space Environ Med. 1988; 59: 399 – 406. ● Rolnick, A., & Lubow, R. E. (1991). Why is the driver rarely motion sick? The role of controllability in motion sickness. Ergonomics, 34, 867– 879. ● Stanney, K., Lawson, B. D., Rokers, B., Dennison, M., Fidopiastis, C., Stoffregen, T., Weech, S., & Fulvio, J. (2020). Identifying causes of and solutions for cybersickness in immersive technology: Reformulation of a research and development agenda. International Journal of Human-Computer Interaction, 36, 1783-1803. https://doi.org/10.1080/10447318.2020.1828535 ● Stoffregen, T. A., Chang, C.-H., Chen, F.-C., & Zeng, W.-J. (2017). Effects of decades of physical driving on body movement and motion sickness during virtual driving. PLOS ONE, 12(11): e0187120. https://doi.org/10.1371/journal.pone.0187120 ● Stoffregen, T. A., & Riccio, G. E. (1988). An ecological theory of orientation and the vestibular system. Psychological Review, 95, 3-14. ● Turner M, Griffin MJ (1999) Motion sickness in public road transport: the relative importance of motion, vision, and 	referenced work about the Postural Instability Theory by Stoffregen and group. We have additionally included references that provide empirical evidence of postural instability preceding motion sickness.
--	--	---

	individual differences. Br J Psychol 90:519–530.	
--	---	--